# Coping Mechanisms and Quality of Life of Low-Income Households during the COVID-19 Pandemic: Empirical Evidence from Bangladesh

**Mohammad Mazharul Islam [1], Mohammad Muzahidul Islam [2,*] and Haitham Khoj [3]**

[1]  Department of Finance, College of Business, King Abdulaziz University, Jeddah 21911, Saudi Arabia
[2]  Department of Management Studies, Patuakhali Science and Technology University, Patuakhali 8602, Bangladesh
[3]  Department of Economics, Faculty of Economics and Business Administration, King Abdulaziz University, Jeddah 21589, Saudi Arabia
*  Correspondence: muzahid@pstu.ac.bd

**Abstract:** It is well known that uncertainty and various measures implemented by the government, such as lockdown, social distancing, and travel restrictions during the COVID-19 pandemic, severely impacted low-income households in Bangladesh. This situation forced them to put forward various mechanisms to cope with the devastating situation caused by the pandemic. This paper focuses on the impact of the COVID-19 epidemic on the quality of life (QoL) of low-income households, their survival coping mechanisms, and the impact of the coping mechanisms on their QoL. From 1st October 2021 to 30th December 2021, primary data from 1279 households were collected through online and offline surveys from different divisions of Bangladesh, and were used to analyze the income-generation, transfer, and cost-minimization practices adopted by the households during the pandemic. The Statistical Package for Social Science (SPSS) version 25 was utilized for data analysis. We employed multivariate and regression statistical techniques to achieve the study objectives. The investigation found that QoL declined significantly due to the COVID-19 crisis. The findings also confirmed that coping mechanisms adopted by households varied according to demographic characteristics, and the QoL deteriorated significantly more in those households that adopted more coping mechanisms relative to others, regardless of socio-demographic features. The findings emphasize the importance of recording grounded survey data to track and gather information on the QoL of low-income households during the pandemic, and of constructing evidence-based policy responses. Furthermore, the study contributes to enriching the existing literature on the impact of the corona pandemic, and can serve as a source for potential studies. This study contributes to a clearer picture of the effects of COVID-19 trauma. This survey-based empirical study provides an understanding of the initial micro-level effects of COVID-19 in Bangladesh. This study gives a synopsis of the extent to which Bangladeshi households adopted mechanisms to deal with the COVID-19 crisis and the effects of the adoption of these mechanisms on quality of life.

**Keywords:** COVID-19; quality of life; coping mechanism; low-income households; Bangladesh

## 1. Introduction

Since December 2019, the globe has faced a comprehensive economic and health disaster due to the COVID-19 pandemic. Many economic sectors are suffering from the consequences of the economic crisis initiated by the COVID-19 pandemic, regardless of the country. The Organization for Economic Co-operation and Development (OECD) stated that the COVID-19 control actions reduced the economic activity of the world, which caused a 50%–100% production loss in certain sectors [1]. This pandemic led to an eco-

nomic crisis expected to increase poverty and wage inequality [2]. Lower-income households will be significantly affected by the COVID-19 pandemic, but the specific effects on these households are still unknown. Numerous studies have been conducted regarding the outbreak of COVID-19 throughout the world [3–6], but most of these studies are focused on the macroeconomic and medical aspects of the COVID-19 crisis. However, limited information is available on socioeconomic and living standard aspects of the COVID-19 pandemic at micro levels [7]. An in-depth evaluation of the impact of COVID-19 at the micro level could produce a significant amount of information to assess the future situation and tackle probable impacts.

The outbreak of COVID-19 has caused an unprecedented economic disaster in Bangladesh, similarly to other countries, with its population significantly vulnerable to income shocks. The pandemic crisis has also adversely affected the well-being of millions of households in Bangladesh. This pandemic has taken a brutal toll on the economy as well as mental health. Due to COVID-19, the average income of all small and medium enterprises (SMEs) and exports fell by 16.93% and 66%, respectively, in 2020 compared to 2019 [8]. According to the Financial Express (2020), about 28 million people have lost their jobs permanently [9]. Regardless of income levels, most people have fallen into an acute financial crisis due to job loss, which has led to the deterioration of their living standards and well-being [10]. At the same time, the drastic reduction in foreign remittance from $2171.03 million in 2020 to $1940.81 million in 2021 due to the pandemic crisis severely affected the standard of living of Bangladeshi households [11]. As a result, the quality of life of a higher number of people declined, with increasing loneliness (71%), depression (38%), anxiety (64%), and sleep disturbance (73%) [12]. A number of studies have denoted the extent of job loss and income decline [10,13]. According to the South Asian Network on Economic Modeling [10], 42% of their surveyed households were found to be below the poverty line. According to the consumer behavior theory, when the incomes of individuals fall, their consumption expenditure does not fall as much, which has been confirmed by the findings of the study conducted by [14] in the Bangladesh case during the pandemic. The wide gap between the income and expenditure behavior of households during the pandemic crisis indicates that there are other strategies, such as income-generation and migration, in addition to expenditure minimizing, that households adopted to cope with the severity of the pandemic crisis. Hence, it is important to focus on the impact of coping mechanisms adopted by households to protect themselves from the COVID-19 pandemic, particularly the impact of these coping mechanisms on the quality of life of low-income households. A deeper and clearer understanding of the potential variations in coping mechanisms across household characteristics and labor market classifications is required in order to construct evidence-based policy responses that can help build a solid and comprehensive social safety net to protect disadvantaged groups from future economic shocks. Therefore, the main objective of this study is to empirically examine the impact of the coping mechanisms adopted during the COVID-19 pandemic on the quality of life (QoL) of households in Bangladesh. To execute the key objective, this study uncovers the socio-economic circumstances of Bangladesh's households experiencing the economic crisis initiated by the corona pandemic. The other objectives are to examine the extent to which the QoL of low-income households was affected by the pandemic, and discover the factors that transformed the QoL of households. Lastly, we recommend policies for a solid and comprehensive social safety net. The rest of the paper is planned as follows. The next section presents the research background in terms of the pandemic crisis and the coping mechanisms and quality of life of low-income households. This section also includes the proposed hypothesis of the study. This is followed by an outline of the study methodology, with some details of research settings, survey questionnaires, data collection techniques, and outcomes presented, including a comprehensive analysis of empirical data and key results. Vital findings are discussed next. Finally, the paper makes conclusions based on important results, and presents practical implications and research limitations.

## 2. Literature Review and Hypothesis Development

### 2.1. Economic Crisis and Quality of Life

An economic crisis is generally seen as a situation in which a country's economy experiences a sudden downturn in overall output or real gross domestic product (GDP). According to Tambunan [15], the effects of an economic crisis are a decrease in real per capita income and an increase in unemployment and poverty. A wholly unprecedented series of emergency lockdowns, compulsory physical isolation, and a temporary course of action implemented by the government and local authorities to block the spread of COVID-19 put household income, employment, health, education, remittances, etc. at high risk [16,17]. The economic crisis was the root cause of several problems for human beings, such as physical and mental health problems [17–19]. For example, Somarriba et al. [20] stated that long-term high unemployment rates and youth unemployment were the main consequences of economic crisis, which certainly has had a severe impact on family and work life. Consequently, the COVID-19 pandemic resulted in a loss in the monthly income of households (including monthly salaries, income from businesses, pensions, leases, bonuses, financial assistance from relatives, etc. [21]. Increased anxiety and depression as well as reduced life satisfaction were found to be linked to economic changes [22]. The economic crisis also impacted household food insecurity [23]. Unpredictable economic changes impact the quality of life due to compromised mental, physical and social well-being [24].

During the pandemic, many people lost their jobs, which increased the unemployment rate, job insecurity, and loss of disposable income, and thus caused a deterioration in the quality of life. QoL is a concept that has been widely used in health care. Quality of life includes , the environment, education, social and religious beliefs economic, and health [25]. Quality of life can be measured by external factors such as the natural environment, political environment, economic environment, social environment, cultural nature, material well-being, social well-being, etc. [26–28]. Internal factors that could be used to measure the quality of life include the individual's feelings and satisfaction with various aspects of physical well-being and personal development [25,29].

People experienced anxiety and stress about managing their finances during the pandemic crisis [30], and poorer households have been adversely affected by spending cuts on essential services [31]. Several types of research conducted during the preliminary phases of the pandemic or lockdown showed that loneliness, restriction of social interactions, and space limitations hurt the mental health and quality of life of children and adolescents [32,33]. It is an axiom that economic capacity plays an integral role in achieving social (sense of belonging, social activities, and affiliation with family or friendship network) and mental well-being (mental health, self-esteem, and life expectancy) [34,35]. QoL along with mental health has been negatively affected in Greece following the 2009 economic crisis, and is only expected to worsen [36]. Therefore, it is argued that any pandemic will induce an economic crisis, leading to a deterioration in the quality of life. As such, this study suggested that low household earners were negatively affected by economic crises during the pandemic period. Therefore, this study proposed the following hypothesis.

**H1:** *The economic crisis caused by the COVID-19 pandemic has negatively affected the quality of life of low-income households in Bangladesh.*

### 2.2. Copying Mechanisms and Quality of Life

During COVID-19, people used various coping mechanisms. Coping is defined as the thoughts and behaviors that enable one to maintain stable emotions, cognitive faculties, behaviors, and physiology during the exhausting time [37]; to manage internal and external burdensome circumstances [38], or to lessen the suffering associated with negative life experiences [39]. On the other hand, coping was also considered to be a technique

an individual utilizes to manage stressors [40]; conscious or unconscious cognitive and behavioral strategies taken by an individual to regulate stress [41,42] (Monat and Lazarus, 1991; Ray et al., 1982), or mechanisms followed by households to survive during unanticipated livelihood failure [43]. Several types of coping strategies are selected by households based on their availability and accessibility. For example, poor households are found to survive shocks by diminishing or changing their expenditure pattern, which could be the most common way for them to survive. Additionally, large households lean towards labor supply by sending a household member to service or augment their own food production. Furthermore, more well-off households have the tendency to utilize their assets to cope with shocks in urban areas [44]. However, Khatri-Chhetri and Maharjan [45] discovered that coping mechanisms hugely influence households' quality of life. The study discovered that poor socioeconomic status and finite resources directly influence the households of the lower classes and underclasses. Higher-educated households have a greater chance of benefiting from stable income sources, and thus are less likely to adopt coping strategies [46]. While multi-generational households do not keep their family members from downgrading their economic well-being. The people least affected during the crisis are the older ones living in one- and two-generation households. They are less likely to utilize these coping strategies, as they can take financial advantage of their pensions [21,47]. Chabowski et al. [47] identified that greater suffering, substandard quality of life, and deprivation are associated with coping based on avoidance strategies. They also noted that coping strategies are employed over the lifetime of an affected person. They depend on several indicators, such as age, sex, socioeconomic status, etc., and vary from individual to individual. Maladaptive coping strategies diminished QoL in the crisis period [48].

Mucci et al. [49] also revealed that the outbreak of COVID-19 had negative impacts on QoL in the general population. Islam and Mostafa [14] conducted a study on the coping strategies of low-income households, and noted that urban return migrants and casual workers have struggled the most in coping with the pandemic, although all income groups have been at risk during the pandemic. The study also released mixed results indicating how male and female respondents utilized coping strategies. Zhan [50] terms 'QoL as the degree to which a person's life experiences are satisfying'. QoL is a multifaceted concept that describes all attributes of a patient's life and well-being. The extremity of the disorder as well as the quality of life may be associated with specific coping mechanisms [51]. Men and women do not have the same choices in terms of quality of life [52]. Lazarus [53] contended that there were no established gender differences in the coping system. While Matheson and Anisman [54] stated that males were most likely to make use of crisis-focused coping, whereas females were feeling-focused coping, since women are interdependent/communal, whereas men are independent/agentic. Rollero et al. [55] investigated that women's QoL was prognostic with social support, while in contrast, men's QoL was more predictive of income level. Quality of life stands on several contextual factors for men and women. The impact on QoL of gender is debatable; for instance, males are inclined to report greater QoL in the physical domain than females, because they experience higher mortality rates and worse life expectancy [56–58]. Women were more likely to take on income-generating strategies, and female-headed households did not indicate a greater reduction in income than their counterparts [8]. Mental health disorders, such as stress, distress, anxiety, etc., reduce the quality of life, and are associated with various socio-economic conditions, such as low level of education, low economic status, unemployment, and suffering, as well as with being female, single, and living alone [59–63].

Dasgupta and Robinson [64] discovered that households headed by women and/or relatively less educated were more affected by COVID-19. These households were more likely to experience food insecurity, as they experienced income loss, while highly educated households were less likely to experience food insecurity, suggesting that education is an essential indicator of QoL. Female-headed households with lower education and lower income level appeared to suffer more food insecurity during this global pandemic.

Older parents (over 39 years) had a greater chance of eluding stressors such as unpredictability, COVID-19 affected information, and were more likely to utilize coping strategies such as communication with others, gardening, and other pastimes [65]. The economic crisis severely affected over 70% of the population aged between 40 and 59 years, and over two-thirds of the people with low incomes in Romania. However, people over 60 were less affected by the economic crisis [66]. Drawing on substantial evidence from the literature, we suggest that coping mechanisms are influenced by various socio-demographic factors that consequently contribute to the quality of life, and we propose the following hypothesis:

**H2:** *Households with higher educated heads might be expected to be less likely to experience coping strategies to sustain their household's quality of life during an epidemic.*

**H3:** *Older adult-headed households might be predicted to need fewer coping mechanisms to sustain their quality of life during outbreaks.*

**H4:** *Households with male heads might be expected to require fewer coping mechanisms to maintain their quality of life during outbreaks.*

**H5**: *Small size households might be predicted to necessitate fewer coping mechanisms to sustain their quality of life during outbreaks.*

**H6**: *Households with a higher number of employed persons might be expected to require fewer coping mechanisms to sustain their quality of life during outbreaks.*

**H7**: *No significant variation is expected among the households across the living region and areas in coping strategies to sustain their household's quality of life during the pandemic.*

**H8**: *The higher the coping mechanisms the larger the decline of quality of life but no significant differences for demographic factors.*

### 3. Method

The systematic approach suggested by Flynn et al. [67] for empirical research was applied in this research. The research gap was articulated through a substantial literature review, and then a survey was performed to examine the objectives.

*3.1. Questionnaire Design and Data Collection Process*

Through a comprehensive literature review, observation instruments were established, pre-tested, and authenticated by a focus group comprising eight persons. The group consisted of one academic specialist from King Abdulaziz University, Jeddah, another academic from Patuakhali Science and Technology University, Patuakhali, Bangladesh, and eight heads of households from eight divisions. The survey was conducted from October 2021 to December 2021. The participants were communicated with via phone and with a cover letter by email simultaneously in order to obtain their consent to participate before the distribution of the study survey. The survey was distributed online as well as offline to 2000 participants by the two expert research assistants, and 1432 responses were gathered after we followed up twice. The respondent was the head of the household. However, based on the usability of the data, the analysis was established on a sample of 1279 (response rate is 63.95%) low-income households headed by individuals aged 18 and older. We conducted a non-response bias test, as our sample was relatively large in size. We split our dataset between two response waves—Wave 1 and Wave 2. The first wave represented the responses received before any reminder was sent to the respondents, and the second wave signified the responses received after a reminder was sent. Wave 2 was treated as a proxy for non-respondents. Then, we used the independent *t*-test to find

whether there were significant differences in our variables across these two new subsample datasets. We found a statistically insignificant difference in variables between the two subsamples where the p-value was more than 0.05. This result suggests that non-response bias was not a concern for this sample. This means that these data should accurately reflect the opinions of low-income households. According to the Japan International Cooperation Agency (JICA) [68], the income level of low-income households in Bangladesh is less than or equal to 31,000 Taka per month. Furthermore, the allocation of respondents was matched with the national population distribution reported by the Bangladesh Bureau of Statistics (BBS) [69] to confirm the representative nature of the respondents (Table 1). For example, the percentages of respondents in this survey obtained from the Dhaka, Chittagong, Rajshahi, Khulna, Rongpur, Barishal, Sylhet, and Maymansing divisions were 25.00%, 20.40%, 12.40%, 10.4%, 9.7%, 6.9%, 7.2%, and 8.1%, respectively. These ratios were very similar to the national population distributions of Dhaka, Chittagong, Rajshahi, Khulna, Rongpur, Barishal, Sylhet, and Maymansing (i.e., 25.3%, 19.7%, 12.8%, 10.9%, 11.0%, 5.8%, 6.9%, and 7.6%, respectively). Thus, it can be claimed that the respondents were representative of the national distribution of populations.

**Table 1.** Sample distribution.

| Variable | Dhaka | Chittagong | Rajshahi | Khulna | Rongpur | Barishal | Sylhet | Maymansing | Total |
|---|---|---|---|---|---|---|---|---|---|
| National Data | 43,417,409 (25.3%) | 33,861,678 (19.7%) | 22,028,304 (12.8%) | 18,686,569 (10.9%) | 18,804,566 (11.0%) | 9,913,505 (5.8%) | 11,797,903 (6.9%) | 13,093,496 (7.6%) | 161,003,430 (100%) |
| Sample | 320 (25.0%) | 261 (20.4%) | 158 (12.4%) | 133 (10.4%) | 124 (9.7%) | 88 (6.9%) | 92 (7.2%) | 103 (8.1%) | 1279 (100%) |
| Residential Area | | | | | | | | | |
| Urban | 239 | 162 | 91 | 75 | 68 | 40 | 48 | 38 | 761 (59.5%) |
| Rural | 35 | 58 | 36 | 35 | 21 | 29 | 26 | 34 | 518 (40.5%) |
| Gender of household head | | | | | | | | | |
| Male | 191 | 134 | 94 | 81 | 73 | 53 | 64 | 66 | 756 (59.1%) |
| Female | 129 | 127 | 64 | 52 | 51 | 35 | 28 | 37 | 523 (40.9%) |
| Family Size | | | | | | | | | |
| Less than 5 | 159 | 115 | 65 | 56 | 57 | 39 | 38 | 43 | 572 (44.7%) |
| 5 to 10 | 98 | 80 | 52 | 38 | 42 | 27 | 31 | 33 | 401 (31.4%) |
| More than 10 | 63 | 66 | 41 | 39 | 25 | 22 | 23 | 27 | 306 (23.9%) |
| Type of house | | | | | | | | | |
| Rented | 163 | 153 | 115 | 95 | 83 | 66 | 54 | 83 | 812 (63.5%) |
| Own | 157 | 108 | 43 | 38 | 41 | 22 | 38 | 20 | 467 (36.5%) |
| Age of household head | | | | | | | | | |
| 18 to 25 | 8 | 7 | 2 | 2 | 1 | 1 | 2 | 7 | 30 (2.3%) |
| 26 to 35 | 31 | 35 | 18 | 14 | 20 | 10 | 8 | 8 | 144 (11.3%) |
| 36 to 45 | 55 | 62 | 36 | 19 | 21 | 19 | 18 | 25 | 255 (19.9%) |
| 46 to 60 | 190 | 124 | 80 | 81 | 59 | 47 | 51 | 57 | 689 (53.9%) |
| More than 60 | 36 | 33 | 22 | 17 | 23 | 11 | 13 | 6 | 161 (12.6%) |
| Type of Occupation | | | | | | | | | |
| Unemployed | 12 | 7 | 6 | 3 | 3 | 9 | 7 | 8 | 55 (4.3%) |
| Self-employed | 117 | 98 | 64 | 60 | 47 | 39 | 28 | 47 | 500 (39.1%) |
| Employee | 181 | 148 | 84 | 68 | 74 | 35 | 48 | 28 | 666 (52.0%) |
| Employer | 10 | 8 | 4 | 2 | 0 | 1 | 3 | 1 | 29 (2.3%) |
| Retired | 8 | 6 | 3 | 3 | 2 | 2 | 3 | 2 | 29 (2.3%) |
| Education levels of household head | | | | | | | | | |
| No education | 28 | 15 | 7 | 11 | 9 | 6 | 0 | 23 | 99 (7.7%) |
| Primary education | 72 | 40 | 17 | 28 | 5 | 31 | 16 | 51 | 260 (20.3%) |
| Secondary education | 34 | 49 | 4 | 8 | 2 | 12 | 42 | 8 | 159 (12.4%) |

| | | | | | | | | | |
|---|---|---|---|---|---|---|---|---|---|
| Diploma | 57 | 58 | 44 | 50 | 32 | 11 | 15 | 12 | 279 (21.8%) |
| Bachelor | 48 | 45 | 32 | 26 | 28 | 10 | 10 | 6 | 205 (16.1%) |
| Master/Ph.D. | 81 | 54 | 54 | 10 | 48 | 18 | 9 | 3 | 277 (21.7%) |
| Number of income earners in the family | | | | | | | | | |
| 1 | 206 | 160 | 114 | 86 | 69 | 63 | 56 | 79 | 833 (65.1%) |
| 2 | 94 | 81 | 36 | 43 | 49 | 20 | 34 | 20 | 377 (29.5%) |
| More than 2 | 20 | 20 | 8 | 4 | 6 | 5 | 2 | 4 | 69 (5.4%) |

### 3.2. Measures of the Constructs

### 3.2.1. Quality of Life

As reported by World Health Organization (WHO), Quality of Life (QoL) stands for 'individuals' perceptions about value systems in which they live and their position in life concerning the culture, goals, expectations, and standards' (WHO) [70]. Many studies linked QoL with various factors, such as health [71–74] and social and economic position [75,76]. A six-item construction of QoL (COV19-QoL) was employed by this study and then justified by Repišti et al. [77]. A 5-point Likert scale was used for rating each item, ranging from 1 (strongly disagree) to 5 (strongly agree), and participants were asked to offer their views for each item. Taking the average of all item scores, a total score was computed, where higher scores indicated a greater impact of COVID-19 on QoL. The factor loading for each item of the QoL scale was >0.70, and the average variance extracted (AVE) for each construct was greater than the highest correlation with any other constructs, which confirmed the construct validity and discriminant validity. The value of Cronbach's alpha (0.813) also confirmed the reliability of the construct (Table 2).

**Table 2.** Results of principal component analysis, reliability, internal consistency, and descriptive statistical values of the COV19-QoL scale in samples.

| Items and Scale | Loadings | Mean | SD | Cronbach's Alpha | AVE | $(r^2)^2$ |
|---|---|---|---|---|---|---|
| Quality of life scale (QoL) | | 3.84 | 0.766 | 0.813 | 0.671 | |
| The quality of our life is lower than before COVID-19. | 0.769 | 4.14 | 0.994 | | | |
| My mental health has deteriorated since COVID-19. | 0.888 | 4.11 | 1.047 | | | |
| My physical health has deteriorated since COVID-19 | 0.719 | 4.11 | 1.048 | | | |
| I feel more tense than before COVID-19. | 0.907 | 4.03 | 1.092 | | | |
| I feel more depressed than before COVID-19. | 0.837 | 3.28 | 1.106 | | | |
| I feel more risk to my personal safety than before COVID-19. | 0.779 | 3.36 | 1.097 | | | |

### 3.2.2. Coping Mechanisms

Combining the existing literature with interview data from three expert educators and six male and two female heads of households from eight divisions, we developed a total of 34 items to measure the 'coping mechanisms' adopted by households to survive the COVID-19 pandemic. All items were assessed using the values of 1 (yes/agree) and 0 (no/disagree), and respondents were requested to indicate their feelings and thoughts for each item on the measure. We employed factor analysis that extracted three factors with a factor loading above 0.50. Based on their nature, we named these three factors the income-generating coping mechanism (IGCM), the expenditure-minimizing coping mechanism (EMCM), and the migration coping mechanism (MCM). The items of the IGCM, EMCM, and MCM scales were acceptable as newly developed items [78], as the factor loading for each item was >0.50, which confirmed construct validity. Cronbach's alpha for

IGCM, EMCM, and MCM was computed to be 0.861, 0.870, and 0.704, which confirmed the reliability of the constructs (Table 3).

**Table 3.** Distribution of sample households stratified by coping mechanisms.

| Coping Mechanisms | Number | | Percentage | |
|---|---|---|---|---|
| | Yes | No | Yes | No |
| Income-generating coping mechanisms (Cronbach's alpha = 0.861) | | | | |
| Take up lower status job (loadings = 0.589) | 412 | 867 | 32.2 | 67.8 |
| Carry out outside activities to raise income (loadings = 0.818) | 608 | 671 | 47.5 | 52.5 |
| Children (below 15) go for jobs or take waged employment (loadings = 0.620) | 453 | 826 | 35.4 | 64.6 |
| Wife/husband go out to work (loadings = 0.794) | 562 | 717 | 43.9 | 56.1 |
| Increase the number of jobs performed (loadings = 0.712) | 529 | 750 | 41.4 | 58.6 |
| Increase total number of hours worked (loadings = 0.642) | 418 | 861 | 32.7 | 67.3 |
| Retired individual goes out to work (loadings = 0.509) | 473 | 806 | 37.0 | 63.0 |
| Borrow money from friends/family/relatives/neighbors (loadings = 0.695) | 270 | 1009 | 21.1 | 78.9 |
| Request a loan or credit from the bank or other financial institutions or moneylenders (loadings = 0.720) | 441 | 838 | 34.5 | 65.5 |
| Rent out part of the house (room) to others (loadings = 0.891) | 364 | 915 | 28.5 | 71.5 |
| Rent out or sell land to others (loadings = 0.772) | 234 | 1045 | 18.3 | 81.7 |
| Rent out/sell/mortgage other properties/assets to others (loadings = 0.911) | 254 | 1025 | 19.9 | 80.1 |
| Withdraw saving/investment (loadings = 0.596) | 530 | 749 | 41.5 | 58.5 |
| Cut down financial contribution to parents or family (loadings = 0.645) | 362 | 917 | 28.3 | 71.7 |
| Expenditure-minimizing coping mechanisms (Cronbach's alpha = 0.870) | | | | |
| Reduce expenses for education by shifting children from private school to public school (loadings = 0.794) | 710 | 569 | 55.5 | 44.5 |
| Stop children from going to school (loadings = 0.611) | 417 | 862 | 32.6 | 67.4 |
| Stop children from pursuing higher education (loadings = 0.639) | 431 | 848 | 33.7 | 66.3 |
| Apply for an education loan (loadings = 0.791) | 679 | 600 | 53.1 | 46.9 |
| Stop paying utility bills (loadings = 0.812) | 513 | 764 | 40.1 | 59.7 |
| Cut down meals (loadings = 0.573) | 790 | 489 | 61.8 | 38.2 |
| Buy cheaper food (loadings = 0.810) | 321 | 956 | 25.1 | 74.7 |
| Stop paying rent (loadings = 0.582) | 282 | 996 | 22.0 | 77.9 |
| Reduce the frequency of meals (loadings = 0.776) | 480 | 799 | 37.5 | 62.5 |
| Cultivate vegetables for self-use (loadings = 0.761) | 358 | 921 | 28.0 | 72.0 |
| Intensify utilization of government health facilities (loadings = 0.512) | 615 | 664 | 48.1 | 51.9 |
| Increase utilization of traditional medicine (loadings = 0.539) | 384 | 895 | 30.0 | 70.0 |
| Cut back visits for treatment in private hospital/clinic (loadings = 0.815) | 397 | 882 | 31.0 | 69.0 |
| Discontinue paying for health assurance (loadings = 0.804) | 494 | 785 | 38.6 | 61.4 |
| Put off purchase of less necessary items (loadings = 0.736) | 514 | 765 | 40.2 | 59.8 |
| Buy local products (loadings = 0.770) | 534 | 745 | 41.8 | 58.2 |
| Renegotiate or stop paying the mortgage (loadings = 0.735) | 350 | 929 | 27.4 | 72.6 |
| Migration (Cronbach's alpha = 0.704) | | | | |
| Migrate to another city or country or to own village (loadings = 0.679) | 551 | 728 | 43.1 | 56.9 |
| Migrate to another area within the municipality (loadings = 0.741) | 291 | 988 | 22.8 | 77.2 |
| Leave rented house and share house with others for free (loadings = 0.758) | 363 | 916 | 28.4 | 71.6 |
| Others (loadings = 0.744) | 262 | 1017 | 20.5 | 79.5 |
| Levels of Coping Mechanisms | | | | |
| Income-generating coping mechanisms | Number | | Percentage | |
| No coping mechanism | 215 | | 16.8 | |
| One or two coping mechanisms | 287 | | 22.4 | |
| Three or more coping mechanisms | 777 | | 60.8 | |
| Expenditure-minimizing coping mechanisms | | | | |
| No coping mechanism | 160 | | 12.5 | |
| One or two coping mechanisms | 132 | | 10.3 | |
| Three or more coping mechanisms | 987 | | 77.2 | |
| Migration | | | | |

| | | |
|---|---|---|
| No coping mechanism | 562 | 44.0 |
| One or two coping mechanisms | 490 | 38.3 |
| Three or more coping mechanisms | 227 | 17.7 |

### 3.2.3. Socio-Demographic Items

We gathered and employed respondents' data for age, gender, residential division, residential area, family size, type of house, educational level, occupation status, and number of income earners.

### 3.3. Study Design

This empirical study applied a survey to test the hypotheses. A multivariate analysis was performed together with a descriptive snapshot. Three logistic regression analyses were performed discretely for three different categories of coping mechanisms to explore which demographic features of respondents most significantly influenced the individual adoption of coping mechanisms. All mechanisms adopted by households in Bangladesh during the COVID-19 pandemic were considered in three categories, namely, the income-generating coping mechanism (IGCM), the expenditure-minimizing coping mechanism (EMCM), and the migration coping mechanism (MCM). The forms of the specific regression models for IGCM, EMCM, and MCM were as follows:

$$IGCM = \beta_0 + \beta_i X_i + \varepsilon_i \tag{1}$$

$$EMCM = \beta_0 + \beta_i X_i + \varepsilon_i \tag{2}$$

$$MCM = \beta_0 + \beta_i X_i + \varepsilon_i \tag{3}$$

Here, the IGCM dependent variable indicated by IGCM = 1 means the household adopted the income-generating coping mechanism during the COVID-19 pandemic crisis period, and IGCM = 0 refers to households that did not adopt it. The dependent variable indicated by EMCM = 1 means the household adopted expenditures coping mechanism during the COVID-19 pandemic crisis period and EMCM = 0 refers to the household that did not adopt it. The MCM dependent variable that indicated by MCM = 1 means the household adopted migration coping mechanisms during the COVID-19 pandemic crisis period, and MCM = 0 refers to households that did not adopt them. $X_i$ is a vector of explanatory variables, and $\varepsilon_i$ refers to the error term.

Further, a regression analysis was then carried out to further investigate coping mechanisms and demographic factors associated with the changes in QoL of low-income households in Bangladesh that caused the COVID-19 crisis. SPSS version 25 was used for data analysis. The specific form of the regression model for QoL is as follows:

$$Y_{QoL} = \beta_0 + \beta_i X_i + \varepsilon_i \tag{4}$$

where $Y_{QoL}$ is the dependent variable that represents the household's QoL, $X_i$ is a vector of explanatory variables, and $\varepsilon_i$ refers to the error term.

## 4. Results Analysis

### 4.1. Profile of Participants

According to Kline's [79] classification, this study's sample size was large enough. Furthermore, the smallest observation-to-variable ratio of 5:1 was obtained from the sample-to-variable ratio model, even though ratios of 15:1 or 20:1 are ideal [80]. Hence, 15 to 20 respondents are strongly recommended to count each and every independent variable in the model for determining the sample sizes. This showed that this rule can also be utilized for multiple regressions and similar analyses. Hence, the adequate sample should be 640. Therefore, it is justified that the sample size of this study was large enough and appropriate for analysis. Furthermore, the study used a bootstrapping method (with n =

500 bootstraps resample). Moreover, Hair et al. [81] confirmed that a response rate of more than 50% is highly suitable for survey-based research.

Table 1 shows that approximately 59% (756) of the respondents were male and 41% (523) were female in the research. The ages of 54% (689) of respondents were between 46 and 60 years, followed by ages between 36 and 45 years (about 20%), more than 60 years (about 13%), between 26 and 35 years (about 11%), and 18 and 25 years (2%). About 59.5% (761) of respondents were from urban areas and 40.5% (518) were from rural areas. Most of the respondents (64%—812) were living in rented houses, while 36% (476) were living in their own houses. About 45% (572) of respondents were living together with less than five family members. Regarding the type of occupation, the majority of the respondents were employees (52%—666), followed by self-employed (39.1%—500), and the remaining 8.9% (113) of the respondents were unemployed, employers, or retired. Most of the participants (65.1%—833) were from single-earner families, followed by those from families with two earners (29.5%—377), and three and more earners (5.4%—69). For education level, 21.8% (279) of the respondents had diploma degrees, 21.7% (277) had postgraduate degrees, 20.3% (260) had primary education, 16.1% (205) had bachelor's degrees, 12.4% (159) had secondary education, and only 7.7% (99) respondents had no education.

### 4.2. Descriptive Statistics

Table 2 represents the nature and grade of six features of QoL together with the total scale. The participants were requested to rank their present levels to show their opinions and feelings about each element on the scale during COVID-19. The results revealed that the corona pandemic largely reduced the QoL of low-income households in Bangladesh by a significant degree (mean = 3.84, SD = 0.766). The life status aspect was reported as the domain most affected by COVID-19, with a mean = 4.14 (SD = 0.994), and the affection domain (feelings, emotions, etc.) was the least affected, with a mean = 3.28 (SD = 1.106). In general, this result offers a satisfactory indication that QoL was negatively affected by the crisis caused by the COVID-19 pandemic, which supports Hypothesis 1 ($H_1$).

Table 3 represents the response of households regarding the adoption of coping mechanisms during the COVID-19 crisis. It shows that at least one IGCM was adopted by more than 83.2% of households, and 60.8% of households applied more than three IGCMs to cope with the severity of the COVID-19 crisis. The highest number of households (47.5%) reported that they carried out outside activities to generate income during the COVID-19 crisis, while the lowest (18.3%) number of households rented out or sold land to others. Regarding expenditures minimizing coping mechanisms, more than 90% of households reported that they implemented at least one mechanism, and most of them (78.5%) executed more than three mechanisms to survive during the pandemic situation. The highest percentage of households (61.8%) stated that they cut down the number of meals to minimize their family expenditure during the pandemic crisis, while the lowest percentage of respondents (22.0%) reported that they stopped paying rent. As reported in Table 3, 55% of households in the survey indicated that they adopted at least one migration strategy as a result of the pandemic to protect them from the harshness of the COVID-19 crisis, and the majority (40.1%) experienced one or two mechanisms. About 28.4% of households migrated to the houses of others and shared accommodation with low or free rent; others migrated to another area within the municipality (22.0%) or migrated to another city or country (13.3%).

### 4.3. Logistic Regression Analysis

The logistic regression analysis of IGCM, EMCM, and MCM was conducted, and the results are presented in Table 4. Households who did not meet any of these strategies during the time of the pandemic were considered as a reference category, in opposition to those households who carried off at least one strategy. There was a strong relationship between household education level and the likelihood of adoption of income-generating coping mechanisms noted from regression analyses. Households with higher-educated

heads had less chance of utilizing these coping mechanisms. For example, to deal with the COVID-19 crisis, households whose head had a diploma, undergraduate, postgraduate, or Ph.D. degree adopted 0.493, 0.365, and 0.418 times fewer income-generation strategies; 0.391, 0.372, and 0.267 times fewer expenditure-minimizing strategies, and 0.553, 0.426, and 0.386 times fewer migration strategies, respectively, compared to households whose heads had no education. In all categories of coping mechanisms, households whose heads were educated at primary and secondary levels did not reveal substantial variations compared to the reference category.

Regarding residential areas, the empirical results suggest that the households of all divisions except the Rajshahi division adopted more mechanisms than households living in the Dhaka division; however, none of the coefficients were statistically significant except for the households who were living in Chittagong. The results also revealed that households of all divisions except Rashahi were less likely to adopt expenditure-minimizing mechanisms compared to the households of the reference division; however, the coefficients were not statistically significant at any level. The consequences of MCM adoption were varied, their coefficients being both positive and negative, but they were not statistically meaningful except in the Chittagong division. Overall, there is no proof that the coping mechanisms practiced by low-income households in Bangladesh are different among residential areas other than the Chittagong division. The households in Chittagong were more 2.559 and 1.593 times more likely to adopt IGCM and MCM than those in Dhaka, where Dhaka was considered a reference residential division.

The results also revealed that urban households adopted MCM 0.696 times more frequently than rural households, while no significant differences were found in terms of IGCM and EMCM. In addition, households with older (46 and above) heads were less likely to adopt IGCM and MCM compared to the reference category, which is statistically significant at ≤0.05 levels. Table 4 also indicates that in all categories of coping mechanisms, there was no significant variation in the gender of household heads, family size, occupation, type of house, and number of income earners, except the households with more than two income earners. Thus, the outcomes of this study did not support Hypothesis 4 and Hypothesis 5, but moderately supported Hypothesis 6, as the study findings revealed that households with more than two income earners were less likely to experience EMCM. These households adopted EMCM 0.508 times less frequently than the reference category.

**Table 4.** Logistic regressions predicting IGCM, EMCM, and MCM.

| Demographic Variables | Model-1 IGCM | | | Model-2 EMCM | | | Model-3 MCM | | |
|---|---|---|---|---|---|---|---|---|---|
| | B | SE | OR | B | SE | OR | B | SE | OR |
| Living Division | | | | | | | | | |
| Dhaka (1) | | | | | | | | | |
| Chittagong (2) | 0.939 ** | 0.235 | 2.559 | −0.680 | 0.459 | 0.507 | 0.466 ** | 0.176 | 1.593 |
| Rajshahi (3) | −0.079 | 0.228 | 0.924 | 0.124 | 0.494 | 1.132 | −0.191 | 0.219 | 0.826 |
| Khulna (4) | 0.184 | 0.254 | 1.202 | −0.289 | 0.496 | 0.749 | 0.044 | 0.223 | 1.045 |
| Rongpur (5) | 0.403 | 0.268 | 1.496 | −0.778 | 0.490 | 0.459 | −0.017 | 0.235 | 0.983 |
| Barishal (6) | 0.014 | 0.295 | 1.015 | −0.323 | 0.517 | 0.724 | 0.041 | 0.259 | 1.042 |
| Sylhet (7) | 0.018 | 0.297 | 1.018 | −0.695 | 0.505 | 0.499 | −0.013 | 0.262 | 0.988 |
| Maymansing (8) | 0.271 | 0.330 | 1.311 | −0.415 | 0.554 | 0.660 | 0.325 | 0.259 | 1.384 |
| Residential Area | | | | | | | | | |
| Urban (1) | | | | | | | | | |
| Rural (2) | 0.016 | 0.156 | 1.016 | 0.232 | 0.197 | 1.261 | −0.362 ** | 0.134 | 0.696 |
| Gender of household head | | | | | | | | | |
| Female (1) | | | | | | | | | |
| Male (2) | 0.270 | 0.142 | 1.310 | 0.065 | 0.177 | 1.067 | 0.154 | 0.122 | 1.166 |

| | BE | SE | OR | BE | SE | OR | BE | SE | OR |
|---|---|---|---|---|---|---|---|---|---|
| **Family size** | | | | | | | | | |
| Less than 5 (1) | | | | | | | | | |
| 5 to 10 (2) | 0.118 | 0.167 | 1.125 | 0.454 | 0.219 | 1.574 | −0.075 | 0.140 | 0.927 |
| more than 10 (3) | 0.010 | 0.178 | 1.010 | −0.088 | 0.211 | 0.915 | −0.005 | 0.150 | 0.995 |
| **Types of houses** | | | | | | | | | |
| Own (1) | | | | | | | | | |
| Rented (2) | 0.202 | 0.158 | 1.224 | 0.196 | 0.195 | 1.217 | 0.041 | 0.133 | 1.042 |
| **Age of household head** | | | | | | | | | |
| 18 to 25 (1) | | | | | | | | | |
| 26 to 35 (2) | −0.124 | 0.671 | 0.884 | 0.275 | 0.696 | 1.317 | −0.370 | 0.432 | 0.691 |
| 36 to 45 (3) | −0.561 | 0.644 | 0.571 | −0.027 | 0.663 | 0.973 | −0.510 | 0.415 | 0.601 |
| 46 to 60 (4) | −0.921 ** | 0.628 | 0.398 | −0.082 | 0.641 | 0.921 | −1.034 ** | 0.402 | 0.355 |
| More than 60 (5) | −0.715 * | 0.656 | 0.489 | 0.048 | 0.677 | 1.049 | −0.932 * | 0.431 | 0.394 |
| **Types of Occupation** | | | | | | | | | |
| Unemployed (1) | | | | | | | | | |
| Self-employed (2) | −0.376 | 0.434 | 0.687 | 0.348 | 0.430 | 1.416 | −0.524 | 0.303 | 0.592 |
| Employee (3) | −0.502 | 0.436 | 0.605 | 0.644 | 0.436 | 1.904 | −0.483 | 0.306 | 0.617 |
| Employer (4) | −0.190 | 0.663 | 0.827 | 1.310 | 0.859 | 3.707 | −0.499 | 0.491 | 0.607 |
| Retired (5) | −0.322 | 0.668 | 0.725 | −0.227 | 0.719 | 0.797 | −0.929 | 0.503 | 0.395 |
| **Education level of household head** | | | | | | | | | |
| No education (1) | | | | | | | | | |
| Primary education (2) | −0.268 | 0.360 | 0.765 | −0.429 | 0.458 | 0.651 | −0.286 | 0.249 | 0.751 |
| Secondary education (3) | −0.531 | 0.394 | 0.588 | −0.036 | 0.537 | 0.964 | −0.353 | 0.280 | 0.703 |
| Diploma (4) | −0.708 * | 0.357 | 0.493 | −0.938 * | 0.452 | 0.391 | −0.593 * | 0.256 | 0.553 |
| Bachelor degree (5) | −1.008 ** | 0.371 | 0.365 | −0.988 * | 0.477 | 0.372 | −0.854 ** | 0.276 | 0.426 |
| Master/Ph.D. degree (6) | −0.872 * | 0.361 | 0.418 | −1.320 ** | 0.453 | 0.267 | −0.953 ** | 0.265 | 0.386 |
| **Number of income earners in the family** | | | | | | | | | |
| 1 (1) | | | | | | | | | |
| 2 (2) | 0.074 | 0.158 | 1.077 | 0.298 | 0.206 | 1.347 | −0.046 | 0.134 | 0.955 |
| More than 2 (3) | 0.091 | 0.336 | 1.096 | −0.678 * | 0.348 | 0.508 | −0.102 | 0.272 | 0.903 |

**Note**: IGCM = Income-generating coping mechanisms; EMCM = expenditure-minimizing coping mechanisms; MCM = migration coping mechanisms; BE = coefficient; OR = odds ratio; SE = standard error; ** $p \leq 0.01$ level, * $p \leq 0.05$ level.

The findings of the regression analysis, which was performed to investigate the association of the coping mechanisms and demographic factors with changes in the QoL during the COVID-19 crisis, are presented in Table 5. From the table, it can be noted that all the coping mechanisms' explanatory variables, such as IGCM, EMCM, and MCM, were significantly related to changes in household QoL. The EMCM was the strongest predictor, followed by IGCM, and MCM. These coping strategies played a key role in the decline of QoL. This result confirmed Hypothesis 8. For example, the results show that the higher the number of cost-cutting strategies, the lower the quality of life (the higher the score of QoL). The coefficients of the variables of 'up to 2' and 'more than 2' for cost-cutting coping mechanisms were found to bear positive signs, with a statistically significant level of 0.01. This result shows that keeping other aspects constant, the higher the households adjusted their consumption behavior to suit the actual or expected reduction in their income and rapidly rising inflation, the lower the QoL. Table 5 also shows that the higher the number of income-generating and migration coping strategies, the lower the quality of life. These coping mechanisms were found to bear positive signs that were statistically significant at

$p \leq 0.05$. This result reveals that with other factors stable, the more the households adjusted to the actual or expected reduction in their income and the rapidly rising inflation through reconfigurations in income arrangements, such as the use of savings or the sale of household goods, etc., the lower the QoL. Among socio-demographic variables, only family size positively and significantly affected the decline in QoL, which also supported Hypothesis 8.

**Table 5.** Results of the multiple regression analysis for changes in household quality of life.

| Variable | Estimated Coefficient (β) | Std. Err. |
|---|---|---|
| Constant | 3.076 (16.869) *** | 0.182 |
| **Income-generating coping mechanism** | | |
| Up to 2 coping mechanisms adopted | 0.153 (2.403) ** | 0.064 |
| 3 and more coping mechanisms adopted | 0.146 (2.490) ** | 0.058 |
| **Expenditure-minimizing coping mechanism** | | |
| Up to 2 coping mechanisms adopted | 0.433 (5.072) *** | 0.085 |
| 3 and more coping mechanisms adopted | 0.946 (12.529) *** | 0.076 |
| **Migration coping mechanism** | | |
| Up to 2 coping mechanisms adopted | 0.08 (0.186) | 0.046 |
| 3 and more coping mechanisms adopted | 0.151 (2.056) ** | 0.073 |
| **Household living region** | | |
| Chittagong | 0.34 (0.585) | 0.059 |
| Rajshahi | 0.34 (0.494) | 0.068 |
| Khulna | 0.72 (1.009) | 0.072 |
| Rongpur | 0.76 (1.020) | 0.074 |
| Barishal | 0.104 (1.232) | 0.084 |
| Sylhet | 0.48 (0.564) | 0.085 |
| Maymansing | 0.005 (0.058) | 0.085 |
| **Number of earners** | | |
| 2 earners | −0.053 (−1.235) | 0.043 |
| more than 2 earners | −0.082 (−0.947) | 0.087 |
| **Family size** | | |
| 5 to 10 members | 0.101 (2.264) ** | 0.045 |
| More than 10 members | 0.070 (1.442) | 0.049 |
| **Residential area** | | |
| Urban residential area | 0.037 (0.891) | 0.041 |
| **Education levels** | | |

| | | |
|---|---|---|
| Primary level | 0.072 (0.872) | 0.083 |
| Secondary level | 0.003 (0.029) | 0.093 |
| Diploma | −0.045 (−0.531) | 0.085 |
| Bachelor | −0.009 (−0.103) | 0.091 |
| Masters/Ph.D. | −0.074 (−0.856) | 0.087 |
| Age of household head | | |
| 26 to 35 years | −0.024 (−0.173) | 0.139 |
| 36 to 45 years | −0.044 (−0.328) | 0.133 |
| 46 to 60 years | −0.093 (−0.717) | 0.129 |
| More than 60 years | −0.169 (−1.222) | 0.138 |
| Gender of head of household | | |
| Female head | 0.041 (1.039) | 0.039 |
| Occupation status of head of household | | |
| Self-employed | −0.099 (−0.994) | 0.039 |
| Employee | −0.069 (−0.684) | 0.039 |
| Employer | −0.194 (−1.218) | 0.039 |
| Retired | 0.018 (0.113) | 0.163 |
| Number of observations | 1279 | |
| d.f | 32 | |
| R² | 0.262 | |
| Adjusted R² | 0.242 | |
| F | 11.742 *** | |

Note: Values in brackets represent the *t*-values of the regression coefficients. *** stands for significant at 0.01 level. ** stands for significant at 0.05 level. The reference categories used in this model are 'no IGCM', 'no EMCM', 'no MCM', Dhaka division, one earner, less than 5 family size, rural area, no education, male head, and informal sector.

## 5. Discussions

It is generally established that the repercussions of the corona pandemic are substantially affecting people's lives, forcing them to adopt strategies to deal with its effects according to their household's socio-economic characteristics. Consequently, the current study attempted to examine not only the effects of such coping mechanisms on quality of life, but also whether these coping mechanisms, as well as their effects, differ according to demographic factors. This investigation is vital to forming a robust, inclusive social safety net, particularly in developing countries. Since the pandemic is not over yet, it could be a long-term cause of fear and anxiety among Bangladeshi low-income households.

Income level has been reduced among all working populations as compared to pre-COVID-19 levels. The COVID-19 pandemic caused an overall 13 percent decline in income levels through the economic and financial crisis in Bangladesh [82]. This decline is greater than those in India and G7 countries [83,84]. The inconsistency could be due to the variation in economic compositions and key markets between countries. Due to a decline in income or lack of income opportunities, households adopted various livelihood-based

coping mechanisms to survive the pandemic situation. A number of studies claim that households in other countries depend on similar coping mechanisms to buffer the effects of sudden income reductions [55,85–88]. Most of these are negative coping mechanisms. The results of this research delivered new understanding of the mechanisms adopted by low-income households in Bangladesh to cope with the consequences of the COVID-19 pandemic. The results of this study confirmed the value of these different mechanisms during the pandemic, which was highlighted in previous studies [21,89]. In addition, this study discovered other measures executed by Bangladeshi households, thus providing a more vivid illustration of the way in which low-income households handle critical circumstances.

A significant finding of this research shows that households living in Chittagong were more likely to carry out income-generating and migration strategies, with such harmful effects as decapitalization (caused by the liquidation of assets, withdrawing savings), indebtedness (resulting from borrowed money from friends/family/relatives/neighbors, loan or credit from banks or other financial institutions or moneylenders), and social capital deterioration. Therefore, the findings of this study did not offer clear support for Hypothesis 7, which stated that there might not be any significant difference among households across the living divisions in adapting mechanisms to maintain their quality of life during the pandemic. This result is consistent with previous studies, which stated that households in different regions included different levels of coping strategies [90,91]. One of the potential explanations for this finding is that they were less likely to implement expenditure-minimizing mechanisms (See Table 4). They tried to maintain their previous expenses on goods and services (i.e., followed pre-COVID-19 consumption patterns) by adopting income-generating and migration mechanisms. Another explanation could be that most of the householders in Chittagong are day laborers and garment workers who have suffered greatly. Moreover, the heads of the remaining households living in Chittagong were mostly transport workers, small traders, hawkers, and small shop owners. These sectors are closely associated with the Garment Industry and Export Processing Zones (EPZs). Prolonged lockdowns and restrictions on various activities destroyed the income sources of these households and at the same time led to inflation, which jeopardized their purchasing power [90,91]. Moreover, after the closure of the garment factory, those who had worked there before were not compensated, and those who depended on small businesses could no longer do so [90,91].

The result of this study further shows that households living in urban areas are more likely to adopt migration strategies than households living in rural areas. This result also did not confirm what was expected by Hypothesis 7. This finding is strongly supported by other studies [92]. There may be different reasons for this result. The first reason may be that the lockdowns and other restrictions imposed by the government during COVID-19 made it difficult for urban households to lead normal lives, with the job losses, reduction in income, and inaccessibility to general services. This situation forced them to return to their villages or other places where the cost of living was low. The second potential reason for this migration from urban to rural could be the fear of spreading the coronavirus, since urban areas are densely populated. The availability of work-from-home opportunities could be another reason for this result. Shutting down all educational institutions for an uncertain period could be another reason for this result.

Another result of this study revealed that households headed by older people were less likely to practice either income-generating or migration coping mechanisms than other households. Thus, this finding confirms what was expected by Hypothesis 3, showing that households with older adult heads needed fewer compatible mechanisms to maintain their quality of life during the pandemic. This finding is also supported by some earlier studies [21,93–95]. It indicates that older people can contribute to supporting family maintenance during the COVID-19 crisis, which, however, does not indicate that households with older heads enjoy more benign economic settings than their counter-

parts. On the contrary, they live in extreme economic uncertainty due to low pension levels. Older people in Bangladesh are facing higher income vulnerability compared to the national average. The consumption of households with older heads mainly depends on transfers and savings, where the monthly transfer value is only BTD 500 [82]. There could be several reasons for this interesting finding. The first possible reason for this result may be that their earnings were secure when the labor market was shrinking, as it was originally received from retirement income, which protected them from the fluctuations of the labor market. Even though the pension money was less, it was a steady income flow for their family, against which they had already adjusted their household expenditure pattern. Another possible reason could be that they retained employment because of their greater work experience and higher work status. The financial support from the government and community for elderly citizens as the targeted group could be another factor protecting them from the need for severe income-generating and expenditure-minimizing coping mechanisms [96]. Another potential reason could be their previous experience of adapting to economic and financial tragedies.

Another result of this study showed that households with higher-educated heads are less likely to carry out income-generating, expenditure-minimizing, and migration strategies to cope with the COVID-19 crisis. This result offers support for Hypothesis 2, which expected households with higher educated heads to be less likely to adopt coping mechanisms to increase income and reduce household expenses and migration compared to other households. This finding is consistent with the finding of previous studies [97]. Adams-Prassl et al. [97] stated that households with lower levels of educational attainment were more vulnerable to the coronavirus pandemic and were more likely to lose their jobs or have their working hours and wages reduced. As a result, these households were more serious about adopting coping strategies compared to households with higher educational qualifications. It is proven that there is a direct significant correlation between education level and job security [98], and educational attainment has a considerable positive impact on one's earnings [84]. Evidence from the literature also suggests that during the COVID-19 pandemic, people who had high job security and more resources were not at risk of exposure to financial difficulties [99].

Another finding of this study was that households with more than two employed persons were less likely to adopt coping mechanisms. This result is consistent with research in other countries that suggests that households with a higher number of employed persons were more likely to offset income risks [21,95]. This finding is explained by the fact that a higher number of earning members in the household results in a higher household monthly income, and the household is thus not at risk of exposure to financial difficulties in maintaining its consumption expenditures. It was therefore not necessary for such households to adopt coping strategies during the COVID-19 crisis. One possible explanation could be that although COVID-19 had a devastating effect on the labor market, the salaried job was less affected. If a household had a higher number of earning members, some of them would be more likely to be salaried earners in government as well as the private sector with jobs that were more secure due to remote working opportunities. Consequently, households with waged employees may have dealt with the effects of the COVID-19-caused interruptions to food systems better than others. The small sample size could be another reason for this result.

We conducted a regression analysis to discover the main relationship between coping strategies, socioeconomic characteristics, and quality of life (QoL) during the COVID-19 crisis. The descriptive analysis showed that the mean score on the COV19-QoL scale was 3.84 (Table 2), demonstrating that the economic crisis caused by the coronavirus outbreak had a substantial negative impact on the quality of life of households in Bangladesh. Hence, the first hypothesis (H$_1$), namely, that the economic crisis caused by the COVID-19 pandemic negatively affected the QoL of low-income households in Bangladesh, is accepted. This outcome validates the results of past studies [77,100–104]. The majority of respondents reported that their social status and the mental and physical health of their

families deteriorated more than before the COVID-19 pandemic. The severity of the impacts of COVID-19 on QoL varies across the country. The potential reasons for different levels of severity could be cultural differences, health level differences, health system differences, and availability of healthcare centers across countries and different study period.

To test the hypotheses about the impact of COVID-19 on the QoL of households in Bangladesh, we performed regression analysis, where the QoL was regressed onto demographic variables (gender, age, educational level, living regions and areas, family size, occupation status, and number of earning family members) and coping mechanisms. The results reported that households living in regions other than Dhaka, with a smaller number of earners, large family size, living in urban areas, lower-educated heads, young heads, female heads, and unemployed heads, experienced greater decline in quality of life by COVID-19 crisis compared to others. However, there was no statistically significant variation among demographic variables except family size (Table 5). Therefore, the eighth hypothesis ($H_8$), namely, that the adverse effects of COVID-19 on QoL of Bangladeshi households did not differ among demographic variables, is accepted. This finding is supported by [100]. This means that households across the country experienced deterioration of QoL during the COVID-19 crisis regardless of the demographic characteristics of the households. There is reason to believe that all households experienced the crisis similarly. The large households experienced significantly more deterioration of QoL than the small households. This variable was found to bear a positive sign and was statistically significant at 0.05 level. This means that with other factors constant, QoL declined more in bigger households during the COVID-19 crisis than in smaller households. The regression coefficient of 2.264 means that with other variables constant, the QoL on average decreased by about 2.264 percent for every one-unit increase in size of a household. The reason for this could be that people were forced to stay at home by the nationwide partial lockdown in Bangladesh, and had no earnings to bear living expenses. Thus, the bigger the household size, the higher the economic burden. This result is strongly consistent with the general economic theory and the findings of the study by [105–107].

To cope with the negative consequences of the COVID-19 crisis, households changed their consumption patterns or sources of financing for their consumption. Among those strategies, the three coping mechanisms that were most widely adopted were IGCM, EMCM, and MCM. Referring to coping mechanisms, our hypothesis H8 was fully confirmed. The result shows that there is a strong positive relationship between coping mechanisms and the decline in QoL. This means that all the categories of coping mechanisms, namely, income-generating coping mechanisms, expenditure-minimizing coping mechanisms, and migration coping mechanisms, significantly increased the deterioration of QoL of Bangladeshi households. The greater the number of coping mechanisms, the lower the quality of life of the households. This finding is supported by other studies regarding issues dealing with a certain stressful situation or traumatic event [55,108–111]. For example, Rollero et al. [55] stated that coping strategies diminish both physical and psychological QoL. Mystakidou et al. [112] clearly state that income loss eventually leads the higher levels of depression, mental complaints, and illness.

Households were forced to adopt IGCMs (decapitalization and indebtedness) due to a sudden sharp decline in their incomes as a result of the long lockdown and other mobility restrictions imposed by the government during the pandemic crisis. These unhealthy income-generating coping mechanisms were found to be a negative significant influence on mental health, socioeconomic conditions, and livelihood, because financial worry is one of the most common stressors in modern life. Financial challenges take an enormous toll on human mental and physical health and on the overall quality of life of a human [110]. Decapitalization and indebtedness lead to deterioration of social status and can cause physical symptoms, such as sleep loss, anxiety, headaches/migraines, compromised immune system, digestive problems, high blood pressure, muscle tension, heart arrhythmia, depression, feelings of overwhelm, low mood, and low energy, resulting in a deterioration of mental and physical health and social conditions. Finally, these psychological and physical sequelae

lower the quality of life of households [113]. The more decapitalization and indebtedness households have, the more likely they are to have poorer quality of life.

This decline in employment, income, market access, and access to services forced families to make significant adjustments in their spending by buying cheap food, stopping rent, bills, and mortgages, and increasing the use of conventional drugs and other measures. Since present earnings are certainly the main source of funds for consumption expenses [114], households were forced to adopt expenditure-minimizing coping mechanisms due to the sharp decline in their incomes, which seems to have contributed to higher levels of food and health insecurity. The reduction in the sizes of consumers' baskets and also in the quality of goods and services contributed to the deterioration of their quality of life. This is congruous with the earlier research and relevant economic theory on QoL, which indicates that consumption and quality of life are positively (though not necessarily linear) associated [115,116]. These unhealthy processes have increased stress, anxiety, and depression, which has led to an overall decline in QoL. The potential reasons for this could be the complex financial portfolio and less or no access to formal health insurance, which may threaten mental health and contribute to lower relative social status and social isolation. Furthermore, low spending on proper health care may increase the burden of acute and chronic health conditions. Reduced physical activity and poor physical health can affect mental health in many ways, such as adding to the financial costs of illness, causing chronic pain and stress about health and mortality, etc. It is also possible that a sharp decline in income leads to social isolation and loneliness and to a decline in social status, which increases frustration and anxiety [117]. Together, these are associated with a lower quality of life [118].

## 6. Practical Implications

Multidisciplinary approaches adopted by low-income families to combat the COVID-19 crisis have negatively affected their quality of life. Based on the findings of this study, several actions were identified as potential strategic guidance for the government in Bangladesh. Thus, we offer suggestions considering the needs of both today and tomorrow in order to avoid the severity of future epidemics, eliminate inequality, and develop and strengthen a solid and comprehensive social safety net. First, there is an urgent need to expand social security programs through increased government support, such as cash transfers to vulnerable households, in addition to current social security spending. Second, as the economy begins to recover, existing policies and solutions should be modified or designed to provide the most comprehensive and prudent solutions. which would increase private demand that could indirectly stimulate supply-side effects, contribute to income enhancement, and create jobs and livelihood opportunities for the households affected by the COVID-19 epidemic. Third, as the decline in income of vulnerable households played a significant role in the affordability and access to healthcare services, the government of Bangladesh should give priority to the health sector in terms of budget allocation for providing proper healthcare. As corruption is one of the main problems in Bangladesh, the government should develop a comprehensive and effective monitoring system to make sure that target group households receive the full benefits offered by the government.

## 7. Conclusions

This survey-based empirical study provides an understanding of the initial micro-level effects of COVID-19 in Bangladesh. This study gives a synopsis of the extent to which Bangladeshi households adopted mechanisms to deal with the COVID-19 crisis and the effects of adopted mechanisms on quality of life. This study found evidence that the severity of the COVID-19 outbreak has significantly reduced household income, forcing them to adopt several mechanisms to deal with the crisis. This study confirmed that the coping mechanisms practiced by households vary according to demographic features, such as region, area, age, number of income earners, and level of education. For example,

young and less educated households, households living in urban areas, households in the Chittagong division, and those with fewer earning members adopted more strategies to cope with the COVID-19 crisis compared to others. The findings of this study also confirmed that the quality of life deteriorated significantly more in those households that adopted more coping mechanisms relative to others, regardless of socio-demographic features. The impacts of the adoption of coping mechanisms on quality of life may be felt for many years to come.

This study has several strong contributions. First, to our knowledge, this is the first time a study has revealed the effects of COVID-19 at the micro-level. Second, this study provides a vivid picture of the impacts of coping mechanisms adopted to deal with the COVID-19 crisis and the effects of these mechanisms on quality of life at the household level. Finally, the results highlight the importance of generating on-the-ground survey data to track household living standards during COVID-19 and gathering the information needed to develop evidence-based policy responses. From this perspective, this study is a pioneer study. It can therefore help policymakers learn what is essential to address immediate, medium- and long-term needs simultaneously.

### 8. Limitations and Future Directions of the Study

Nonetheless, this research has several limitations. The main limitation of this study is that all respondents were of lower income, thus limiting the generalizability of the results. Another limitation of the study is that effects on quality of life based on single-period cross-sectional data may not be justifiable due to the multifaceted character of the COVID-19 pandemic. A randomized future study may be more acceptable to establish association and causation. In addition, we also recommend future investigations to explore a comprehensive link between coping mechanisms, quality of life, and epidemic crisis by taking into account all other elements of quality of life. This study provides greater knowledge of their primary impact.

**Author Contributions:** M.M.I. (Mohammad Mazharul Islam) conceptualized, designed, analyzed data, and drafted the manuscript. M.M.I. (Mohammad Muzahidul Islam) collected data, and drafted and revised the manuscript. H.K. drafted and revised the manuscript. All authors have read and agreed to the published version of the manuscript.

**Funding:** This study involved no external funding.

**Institutional Review Board Statement:** Not applicable.

**Informed Consent Statement:** We revised Ethics approval and consent to participate into IRBS.

**Data Availability Statement:** The datasets used and analyzed during the current study are not openly available due to the sensitivity of the personal data, and are available from the corresponding author at a reasonable request.

**Acknowledgments:** We express noble gratitude to all the participants who participated in the survey. We are also grateful to our research assistants, who were involved in data collection. I also acknowledge the great contribution of Muzahidul Islam and Haitham Khoj.

**Conflicts of Interest:** The authors declare that they have no competing interests.

### Abbreviations

QoL: quality of life, OECD: the organization for economic co-operation and development, IGCM: income-generating coping mechanism, EMCM: expenditure-minimizing coping mechanism, MCM: migration coping mechanism.

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
