# Peer review of "Coping Mechanisms and Quality of Life of Low-Income Households during the COVID-19 Pandemic: Empirical Evidence from Bangladesh"

_sustainability, doi:10.3390/su142416570_

Round 1

Reviewer 1 Report

This is an interesting topic and a well-written paper. The authors have done extensive research. Hypotheses are clearly stated. The methodology seems to be sound. The article offers important policy implications. Some minor spellcheck is required. 

The study provided an understanding of the micro-level effects of COVID-19 in Bangladesh.The topic is original in that it uses primary data of 1279 households. There are not many published studies in the micro level effects of COVID-19. The main contribution of the paper is the emphasis of the research on the family level and QoL.The main limitation of the research is that it is concentrated on lower income population. Conclusions are very extensive and summarize all the results of the survey. A large number of references is acknowledged.

Reviewer 2 Report

It’s my pleasure to read this fantastic manuscript entitled “Coping Mechanisms and Quality of Life of Low-Income Households During the COVID-19 Pandemic: Empirical evidence from Bangladesh. The way authors approached the paper is amazing. I believe that this manuscript can be a good candidate for publication. The

following comments for the reference for authors:

1) The title is great. The abstract is ok. Authors may put the sampling technique and the

name of statistical tools (software) used in the abstract will be great.

2) Authors may add one or two extra paragraphs regarding the COVID-19 scenarios in

Bangladesh during the pandemic, especially the living standard of high income

society and low income society. Authors may add the objective of the study at the last

para of introduction. Authors should split the first para of introduction as it is bit long.

Authors may add the organization of the manuscript at the end of introduction.

Authors should add more recent relevant papers in the introduction section (for

example: 2018-2022)

3) In the “Review of Literature” section of 2.1. Economic Crisis and Quality of Life”

authors should cite more recent papers. The following lines deserve citations I think:

“Particularly unprecedented series of emergency lockdowns, compulsory physical

isolation, a temporary course of action implemented by the government and local

authorities to block the spread of COVID-19 put the households’ income,

employment, health, education, and remittances, etc. at extremely high risk. The

economic crisis is the root cause of several problems for human beings such as

physical and mental health.”

“2. Review of Literature” can be renamed as “Literature Review and Hypotheses

Development”

4) The “3. Method” section is ok. Have authors checked the non-response bias as the

sample size is comparatively higher? If yes, they may mention it in the revised

version.

5) The Discussion and Conclusion is ok.

6) Authors mat separate the “Implication of the study” from the conclusion. Also,

authors should put the “Limitations and Future Directions of the study” for the

reference of future researchers.

7) Authors should check the tables and figures numbers, in-text citations and references for more accurate matching.

8) My last concern is about the research contribution to Coping Mechanisms and Quality of Life literature. I recommend the authors to discuss deeply the paper contributions in the introduction section and also at the end of paper. 
